# An Integrated PMA Pretreatment Instrument for Simultaneous Quantitative Detection of *Vibrio parahaemolyticus* and *Vibrio cholerae* in Aquatic Products

**DOI:** 10.3390/foods14132166

**Published:** 2025-06-21

**Authors:** Yulong Qin, Rongrong Xiong, Yong Zhao, Zhaohuan Zhang, Yachang Yin

**Affiliations:** 1College of Food Science and Technology, Shanghai Ocean University, Shanghai 201306, China; qyl1218@139.com (Y.Q.); rongr_x@163.com (R.X.); yzhao@shou.edu.cn (Y.Z.); 2International Research Center for Food and Health, Shanghai Ocean University, Shanghai 201306, China; 3Shanghai Hi-Chain Food Co., Ltd., Shanghai 201913, China

**Keywords:** PMA pretreatment instrument, duplex qPCR, *Vibrio parahaemolyticus*, *Vibrio cholerae*, aquatic products

## Abstract

Traditional hazard identification techniques for *Vibrio parahaemolyticus* often neglect the distinction between viable and nonviable bacteria in aquatic products, leading to overestimated disease risks and uncertainties in risk assessments. To address this limitation, we developed an automated PMA pretreatment instrument that integrates dark incubation and photo-crosslinking into a unified workflow, allowing customizable parameters such as incubation time, light exposure duration, and mixing speed while maintaining stable temperatures (<±1 °C fluctuation) to preserve bacterial DNA integrity. Leveraging this system, a duplex qPCR assay was optimized for simultaneous quantitative detection of *V. parahaemolyticus* and *V. cholerae* in aquatic products and environmental samples. The assay demonstrated robust performance with 90–110% amplification efficiencies across diverse matrices, achieving low limits of detection (LODs) of 10^1^–10^2^ CFU/mL in shrimp farming environment water and 10^2^–10^3^ CFU/g in shrimp (*Litopenaeus vannamei*) and oyster (*Crassostrea gigas*). Notably, it effectively discriminated viable bacteria from 10^6^ CFU/mL(g) nonviable cells and showed strong correlation with ISO-standard methods in real-world sample validation. This integrated platform offers a rapid, automated solution for accurate viable bacterial quantification, with significant implications for food safety, pathogen surveillance, and risk management in aquatic industries.

## 1. Introduction

*Vibrio* species are the most common pathogenic bacteria found in seafood, with *Vibrio parahaemolyticus* being the leading seafood-borne pathogen in China. Food poisoning incidents caused by this bacterium rank first among bacterial foodborne illnesses and are considered one of the primary causes of diarrheal diseases worldwide [1,2,3,4]. *Vibrio cholerae*, the main causative agent of cholera, is widely distributed and classified as a severe infectious disease. It has repeatedly caused large-scale outbreaks, primarily characterized by severe vomiting, diarrhea, and dehydration, which can lead to death in severe cases [5,6,7]. Therefore, developing a rapid, accurate, and efficient method for identifying pathogenic *Vibrio* in seafood is crucial for enhancing seafood quality and improving public health, holding significant scientific and practical value.

Traditional methods for quantitatively detecting these two foodborne pathogens in food require approximately one week [8,9,10], including sample collection and pretreatment (Step 1), selective cultivation (Step 2), strain purification (Step 3), physiological and biochemical identification (Step 4), and molecular biological identification (Step 5). Enzyme-linked immunosorbent assay (ELISA) based on immunological techniques can specifically bind and detect the thermostable direct hemolysin (TDH) and TDH-related hemolysin (TRH) produced by pathogenic *V. parahaemolyticus* [11]. However, it cannot directly detect viable bacteria in the samples, which limits its ability to accurately measure the contamination risk in the samples. Lateral flow assays are widely used for on-site detection [12]; however, their limitations are also evident, as they can only perform qualitative tests and cannot provide precise quantitative measurements of samples. Although DNA-based quantitative PCR (qPCR) technology is convenient and fast, it cannot distinguish between viable and nonviable bacteria in samples, leading to potential false-positive results [13,14,15]. Propidium monoazide (PMA) is a photosensitive nucleic acid dye that, when combined with qPCR technology, can selectively identify viable bacterial DNA in food, thereby achieving the goal of quantifying viable bacteria [16,17,18]. The PMA pretreatment process involves two key steps: dark treatment and photolysis. The specific operational steps are as follows: First, an appropriate concentration of PMA is added to the sample. Under dark conditions, this dye selectively enters the cells of nonviable bacteria with damaged membranes. Upon exposure to activating light (halogen lamp or LED blue light), the azide groups of PMA molecules have the capability to react with hydrocarbons of double-stranded DNA, forming stable covalent nitrogen–carbon bonds and producing stable covalent crosslinked precipitates. This process inhibits the effective amplification of nonviable bacterial DNA, eliminating the possibility of false-positive results caused by nonviable bacterial DNA, while having no effect on intact viable bacteria [17,19,20].

Traditional PMA pretreatment methods typically use a closed environment for dark treatment, followed by photolysis with a handheld sodium tungsten lamp of approximately 600 W [21,22]. While this method is effective, it is cumbersome, requires manual operation, and is time-consuming. A PMA manufacturer has developed a blue LED tube-based commercial PMA treatment device that significantly simplifies the PMA pretreatment process; however, this instrument has notable drawbacks: it separates the dark treatment and photolysis steps, necessitating cumbersome manual transfer operations during the PMA treatment process, which does not align with the modern design concept of fully automated high-tech instruments.

This study addresses this limitation by constructing a novel instrument for the rapid screening of viable bacterial DNA suitable for PMA-qPCR pretreatment. This instrument integrates the two core steps of dark treatment and photolysis into one, achieving an integrated and fully automated PMA pretreatment process, thereby simplifying the PMA-qPCR pretreatment process from the outset and enhancing PMA treatment efficiency. Based on this integrated PMA pretreatment instrument, we have developed a dual qPCR technology combining the advantages of PMA and multiplex qPCR, specifically targeting *V*. *parahaemolyticus* and *V*. *cholerae* in seafood. This technology can be used for the quantitative detection of viable pathogenic *Vibrio* cells in seafood and related aquatic environments, providing a new detection and analysis tool for identifying the hazards posed by *V*. *parahaemolyticus*.

## 2. Materials and Methods

### 2.1. Materials for PMA Pretreatment Instrument

The LED blue light lamps used in this study were purchased from CREE, USA, totaling 24 units, with a power range of 1–3 W, voltage parameters of 3.0–3.7 V, and a current range of 350–1000 mA. The color temperature wavelength was 410–420 nm, producing blue light with a purplish/deep blue tint. The digital timer was obtained from Wenzhou Dahua Instrument Co., Ltd., Zhejiang, China; the digital rotary regulator was sourced from JSCC, Taiwan, China; the transformer was purchased from Mingwei Co., Ltd., Taiwan, China; and the welding equipment utilized was an argon arc welder from Shenzhen Ruilong Welding Machine, Guangdong, China, model TIG 400GT. The stainless steel plates used for constructing the apparatus were acquired from Xiamen Tungsten Industry Co., Ltd., Fujian, China, measuring 1.5 mm and made of food-grade 304 stainless steel.

### 2.2. V. parahaemolyticus and V. cholerae Strains

The strains used in this study were *V*. *parahaemolyticus* ATCC33847 and *V*. *cholerae* GIM1.449. The GIM1.449 strain was obtained from the Guangdong Institute of Microbiology and is a low-toxicity strain that is non-O1 and non-O139, suitable for routine laboratory research. The glycerol stock of the strains was streaked onto a Chromogenic *Vibrio* agar plate and single colonies were picked and inoculated into 10 mL TSB (1% NaCl, pH 8.0). All experiments were conducted in a biosafety cabinet. The inoculated bacterial suspension was placed in a bacterial incubator at 37 °C with a shaking speed of 200 r/min and incubated overnight for 16–24 h, yielding a *V*. *parahaemolyticus* culture with a concentration of 10^9^ CFU/mL and a *V*. *cholerae* culture with a concentration of 10^7^ CFU/mL.

### 2.3. Design of the Integrated PMA Pretreatment Instrument

The design schematic for the integrated PMA pretreatment instrument, suitable for PMA-qPCR preprocessing, was created using drawing software, as shown in Figure 1. The instrument primarily consists of a PMA top cover, LED blue light source, digital timer, digital rotary regulator, rotary mixing device, and motor chamber. The overall shape of the instrument is rectangular, with approximate dimensions of 25 × 20 × 20 cm. All conceptual diagrams presented in this chapter were drafted using AutoCAD 2016 (Computer Aided Drafting) software, which was purchased from Autodesk, Denver, CO, USA.

### 2.4. PMA Pretreatment Process Based on Integrated PMA Pretreatment Instrument

To prepare a 2 mM PMA stock solution, 980 μL of double-distilled water was added to 1 mg of PMA (Biotium, Hayward, CA, USA). A total of 12.5 μL of the PMA stock solution was then added to the sample to achieve a final concentration of 50 μM PMA in the mixture [23]. The sample was placed in the integrated PMA pretreatment instrument constructed for this study, which was adjusted from our laboratory’s previous research [24,25], with a dark treatment time set to 30 min and a light treatment time also set to 30 min. The instrument was then activated. At a rotation speed of 1000 rpm, the nonviable bacterial DNA in the sample reacted fully with PMA in the instrument, forming a stable covalent crosslinked precipitate to facilitate the selection of viable bacterial DNA. The reacted sample was then centrifuged at 12,000 rpm to collect the viable bacterial cells. The viable bacterial DNA was extracted from the samples using a bacterial DNA extraction kit (Tiangen Bictech (Beijing) Co.,Ltd. Beijing, China. DNA concentration was determined using a NanoDrop ND-2000 spectrophotometer (Thermo Fisher Scientific (Shanghai) Co., Ltd., Shanghai, China).

### 2.5. Primers, Probes, and PCR Reaction Conditions

The primers and probes used for the quantitative detection of *V*. *parahaemolyticus* and *V*. *cholerae* are listed in Table 1 and were custom ordered from Invitrogen, a subsidiary of Thermo Fisher Scientific.

qPCR reaction conditions: predenaturation temperature: 95 °C for 2 min; denaturation at 95 °C for 15 s; annealing temperature: 60 °C for 1 min. The denaturation and annealing processes were repeated for a total of 40 cycles; the melting curve was automatically generated by the system. Reactions were performed in triplicate and average values were taken.

The 20 μL qPCR reaction system consisted of 0.2 μL Taq DNA polymerase, 0.5 μL forward primer, 0.5 μL reverse primer, 0.2 μL TaqMan probe, 2 μL PCR reaction solution, 1.2 μL Mg^2+^ solution, 0.8 μL dNTPs solution, and 1 μL DNA template, with the remainder filled with ddH_2_O to a total volume of 20 μL. For the no template control (NTC), 1 μL of ddH_2_O was added in place of the DNA template. The standard strains *V. parahaemolyticus* ATCC33847 and *V. cholerae* GIM1.449 were used as amplification controls. All reagents were sourced from Invitrogen, a subsidiary of Thermo Fisher Scientific, and the PCR reactions were conducted using the 7500 Fast real-time PCR instrument (Thermo Fisher Scientific (Shanghai) Co., Ltd., Shanghai, China). The postprocessing software for qPCR reactions was 7500 Software v2.0.6, downloaded from the official website of Applied Biosystems, Inc., USA. The C_T_ threshold was set to 10 times the baseline of the fluorescence signal.

### 2.6. Specificity Analysis of PMA Duplex qPCR Technology

The strains used for the specificity detection of PMA duplex qPCR are shown in Table 2, which includes 62 strains of *V. parahaemolyticus*, 31 strains of *V. cholerae*, 3 other strains of the *Vibrio* genus, and 7 non-*Vibrio* strains. DNA from different strains was extracted using a bacterial DNA extraction kit and PMA duplex qPCR detection was performed to validate the specificity and selectivity of the PMA duplex qPCR, with each experimental group repeated three times.

### 2.7. Construction of the Standard Curve for PMA Duplex qPCR Technology

The standard curve for this study was constructed based on actual samples collected from the shrimp farming environment in Fengxian District, Shanghai, where shrimp and oyster were purchased from a seafood market in Pudong New District, Shanghai. A total of 25 g of shrimp and oyster was homogenized in 225 mL of alkaline peptone water (APW) to obtain homogenates. A bacterial suspension of 10^9^ CFU/mL of *V*. *parahaemolyticus* and 10^7^ CFU/mL of *V*. *cholerae* was diluted using the shrimp farming environment water, homogenate of South American white shrimp, and homogenate of oyster to obtain samples containing 10^1^ to 10^8^ CFU/mL(g) of *V*. *parahaemolyticus* and 10^1^ to 10^6^ CFU/mL(g) of *V*. *cholerae*, with each experimental group repeated three times.

PMA pretreatment, DNA extraction, and PMA duplex qPCR quantitative detection were performed, using the results from colony counts and the C_T_ values from duplex qPCR to construct the standard curve for the reaction, thereby determining the limit of detection (LOD) for the PMA duplex qPCR. The limit of quantification (LOQ) was set to 3 times the LOD. All the measurements were conducted in triplicate. The C_T_ values were expressed as the mean ± standard deviation (SD). The amplification efficiency of the PMA duplex qPCR reaction was calculated using the slope (k) of the standard curve with Formula (1)(1)E = 10^−1/k^ − 1,

### 2.8. Preparation of Inactivated Pathogenic Vibrio

A total of 10 mL of 10^7^ CFU/mL suspensions of *V*. *parahaemolyticus* and *V*. *cholerae* were subjected to high-temperature and high-pressure sterilization at 121 °C for 20 min to obtain inactivated bacterial cells. The vitality of the bacteria was confirmed using a colorimetric culture method for *Vibrio* and further validated using tryptic soy broth (TSB) and tryptic soy agar (TSA) through secondary confirmation, incubated at 37 °C for 18–24 h, with no bacterial growth indicating complete bacterial death.

### 2.9. Application of PMA Duplex qPCR in Artificially Inoculated Samples

Fresh shrimp and oyster used for artificial inoculation [26,27] were purchased from a seafood market in Pudong New District, Shanghai, while the shrimp farming environment water was collected from Jixian Shrimp Farming in Fengxian District, Shanghai. A total of 27 groups of artificially inoculated seafood samples were selected, including 9 groups of water environment samples, 9 groups of shrimp, and 9 groups of oysters. These samples were placed in sterile homogenization bags, inoculated with varying proportions of viable and nonviable bacteria, and then thoroughly mixed. Next, we waited 30 min to allow the bacterial solution to evenly adhere to the surface of the sample. A concentration of 10^6^ CFU/mL(g) of nonviable bacteria was added to assess the ability of the PMA duplex qPCR method to distinguish between viable and nonviable bacteria using traditional plate counting methods and standard qPCR methods as controls.

### 2.10. Application of PMA Duplex qPCR in Actual Aquatic Samples

Sampling was conducted at the Dongfang International Aquatic Center in Shanghai, where a total of 108 aquatic product samples were collected, including 48 shrimp samples, 23 shellfish samples, 18 freshwater fish samples, and 19 algae samples. The PMA duplex qPCR technology was employed for a comprehensive survey, validated using ISO standard methods (2007) [26].

### 2.11. Statistical Analysis

All the measurements in this study were conducted in triplicate. The values in this study were expressed as the mean ± standard deviation (SD). The data were analyzed with SPSS statistic 27 software (IBM Co., Chicago, IL, USA). The differences among the mean values were determined by one-way ANOVA. Differences were considered significant at *p* < 0.05. Diagrams were performed by GraphPad Prism 10 (GraphPad Software, San Diego, CA, USA).

## 3. Results and Discussion

### 3.1. Construction of Integrated PMA Pretreatment Instrument

The prototype of the integrated PMA pretreatment instrument constructed in this study is shown in Figure 2. Through practical application, this instrument can autonomously set the dark treatment time, light treatment time, and mixing speed according to the experimental requirements, making it applicable for various PMA pretreatment conditions. Actual measurements using a temperature detector indicated that the cavity temperature did not show significant changes during the continuous use of the instrument for 6 h (Figure 3). All the measurements were conducted in triplicate.

Traditional PMA pretreatment steps typically rely on high-wattage halogen lamps (600 W–1000 W) for photo-linking (Figure 4a). While this method of photo-linking is highly reliable, halogen lamps release a substantial amount of heat during irradiation [22]. This not only risks damaging the DNA of viable bacteria in the samples, leading to false-negative results, but excessive heat can also pose laboratory safety hazards. In contrast, LED blue light serves as an energy-efficient and environmentally friendly cold light source, maintaining low thermal output even during prolonged operation [28,29]. Furthermore, the wavelength of LED blue light, ranging from 410–420 nm, aligns with the excitation wavelength of PMA, allowing for effective crosslinking of PMA with nonviable bacterial DNA without damaging the DNA of viable bacteria [28,29].

Compared to the open LED blue light crosslinking instrument developed by Biotium Company (Figure 4b), the integrated PMA pretreatment instrument constructed in this study integrates the two core steps of dark treatment and photo-linking into one, achieving an integrated and fully automated PMA pretreatment process. This simplification at the source enhances the efficiency of PMA processing and provides reliable tool support for the quantification of viable bacteria.

### 3.2. Specificity Analysis of PMA Duplex qPCR

This study utilized 103 strains of different bacterial species (Table 2) to validate the specificity of the method, with results presented in Table 3: 62 strains of *V*. *parahaemolyticus* showed positive FAM fluorescence signals in qPCR detection, while 31 strains of *V*. *cholerae* exhibited positive Cy3 fluorescence signals, demonstrating that the constructed PMA duplex qPCR technology possesses high specificity. The remaining three strains of other *Vibrio* species and seven strains of non-*Vibrio* species yielded negative results in PMA-qPCR detection, indicating the method’s exceptional specificity. The detection rate was 100%.

### 3.3. Amplification Efficiency of PMA Duplex qPCR

Standard curves were constructed using actual samples to determine the amplification efficiency and quantification limits of the PMA duplex qPCR developed in this study, as shown in Figure 5. Over 40 cycles, none of the negative controls detected a fluorescence signal. In the shrimp farming environment water, shrimp (*Litopenaeus vannamei*), and oyster (*Crassostrea gigas*), the R^2^ values for the standard curve of *V*. *parahaemolyticus* were 0.988, 0.980, and 0.999, respectively, demonstrating a good linear relationship between colony counts and C_T_ values, with slopes of −3.49, −3.16, and −3.47, and amplification efficiencies of 93.43%, 107.23%, and 94.17%. The R^2^ values for the standard curve of *V*. cholerae were 0.994, 0.985, and 0.975, with slopes of −3.49, −3.16, and −3.47, and amplification efficiencies of 90.94%, 94.92%, and 93.07%. The amplification efficiency remained between 90 and 110%, indicating that the PMA duplex qPCR exhibits good amplification efficiency. This study selected complex samples such as water from shrimp farming environments, shrimps, and oysters to construct the standard curve for the PMA duplex qPCR technology, demonstrating that the method maintains high amplification efficiency, showcasing its potential for application in real samples.

### 3.4. Minimum Quantification Limit of PMA Duplex qPCR

Based on the analysis of the minimum quantification limit from the standard curve, the PMA duplex qPCR technology developed in this study achieved a minimum quantification limit of 10^1^–10^2^ CFU/mL in the water of shrimp farming environments, and 10^2^–10^3^ CFU/g in shrimp and oyster. In comparison to traditional PMA pretreatment qPCR quantification techniques, such as the PMA-qPCR method developed by Liu et al. [30], which can detect viable *Escherichia coli* with a detection limit of 10^3^ CFU/mL in pure culture and 10^5^ CFU/g in beef, and the PMA-qPCR system for quantifying viable Photobacterium developed by Macé and Mamlouk [31], which has a detection limit of 10^3^ CFU/g in fresh salmon, the aforementioned methods can only detect one type of bacteria. In contrast, the method designed in this study can simultaneously quantify the two most common and harmful pathogenic *Vibrio* in seafood, exhibiting high sensitivity.

### 3.5. Study on the Differentiation of Viable and Nonviable Bacteria in Artificially Inoculated Samples Using PMA Dual qPCR

In this study, 27 groups of samples were selected, including 9 groups of shrimp farming environment water samples, 9 groups of shrimps, and 9 groups of oysters, inoculated with varying ratios of viable and nonviable bacteria to evaluate the capability of the PMA dual qPCR method in distinguishing between viable and nonviable bacteria in aquatic products. As shown in Table 4, Table 5 and Table 6, in the presence of 10^6^ CFU/mL(g) nonviable bacterial DNA, traditional qPCR technology struggled to differentiate between viable and nonviable bacteria in aquatic products and their associated shrimp farming environment water samples. The quantitative results were significantly higher than those obtained using the PMA dual qPCR technique, which was severely interfered with nonviable bacterial DNA, leading to erroneous estimations of pathogenic bacteria risk in aquatic products [10,11,14,15]. In contrast, the PMA dual qPCR technology developed in this study maintained extremely accurate quantitative results even in complex samples containing both viable and nonviable pathogenic *Vibrio* species: it was able to accurately quantify the viable cells of *V*. *parahaemolyticus* and *V*. *cholerae* in the presence of 10^6^ CFU/mL(g) nonviable bacteria. This indicates that the method effectively avoids the amplification of nonviable bacterial DNA from *V*. *parahaemolyticus* and *V*. *cholerae* in artificially inoculated aquatic products, achieving the goal of quantifying viable bacteria. It is a potential quantitative assessment method for viable bacteria that can be used to accurately assess the true risk of pathogenic *Vibrio* in aquatic products.

**Table 4 foods-14-02166-t004:** The ability of PMA-qPCR to distinguish the nonviable and viable *Vibrio* cells in shrimp farming environment water.

Number of Infections	PMA	qPCR Quantification Results
10^6^ VP nonviable + 10^4^ VP viable	−	5.96 ± 0.83 CFU/mL (VP)
	+	4.23 ± 0.25 CFU/mL (VP) *
10^6^ VP nonviable + 10^3^ VP viable	−	6.13 ± 0.03 CFU/mL (VP)
	+	2.98 ± 0.15 CFU/mL (VP) *
10^6^ VP nonviable + 10^2^ VP viable	−	6.47 ± 0.19 CFU/mL (VP)
	+	2.17 ± 0.37 CFU/mL (VC) *
10^6^ VC nonviable + 10^4^ VC viable	−	6.28 ± 0.53 CFU/mL (VC)
	+	4.16 ± 0.61 CFU/mL (VC) *
10^6^ VC nonviable + 10^3^ VC viable	−	5.79 ± 0.38 CFU/mL (VC)
	+	3.18 ± 0.95 CFU/mL (VC) *
10^6^ VC nonviable + 10^2^ VC viable	−	6.12 ± 0.36 CFU/mL (VC)
	+	1.98 ± 0.62 CFU/mL (VC) *
10^6^ VP nonviable + 10^4^ VP viable + 10^6^ VC nonviable + 10^4^ VC viable	−	6.35 ± 0.75 CFU/mL (VP)
		6.51 ± 0.31 CFU/mL (VC)
	+	3.61 ± 0.16 CFU/mL (VP) *
		3.72 ± 0.86 CFU/mL (VC) *
10^6^ VP nonviable + 10^3^ VP viable + 10^6^ VC nonviable + 10^3^ VC viable	−	6.45 ± 0.88 CFU/mL (VP)
		5.88 ± 0.37 CFU/mL (VC)
	+	2.89 ± 0.63 CFU/mL (VP) *
		3.38 ± 0.95 CFU/mL (VC) *
10^6^ VP nonviable + 10^2^ VP viable + 10^6^ VC nonviable + 10^2^ VC viable	−	5.81 ± 0.68 CFU/mL (VP)
		6.71 ± 0.27 CFU/mL (VC)
	+	2.28 ± 0.76 CFU/mL (VP) *
		2.42 ± 0.69 CFU/mL (VC) *

**Note**: VP: *V. parahaemolyticus; VC: V. cholerae; *: p <* 0.05.

**Table 5 foods-14-02166-t005:** The ability of PMA-qPCR to distinguish the nonviable and viable *Vibrio* cells in shrimps.

Number of Infections	PMA	qPCR Quantification Results
10^6^ VP nonviable + 10^4^ VP viable	−	6.34 ± 0.45 CFU/g (VP)
	+	4.23 ± 0.36 CFU/g (VP) *
10^6^ VP nonviable + 10^3^ VP viable	−	5.82 ± 0.61 CFU/g (VP)
	+	3.19 ± 0.44 CFU/g (VP) *
10^6^ VP nonviable + 10^2^ VP viable	−	6.17 ± 0.59 CFU/g (VP)
	+	2.37 ± 0.37 CFU/g (VP) *
10^6^ VC nonviable + 10^4^ VC viable	−	6.27 ± 0.23 CFU/g (VC)
	+	4.63 ± 0.34 CFU/g (VC) *
10^6^ VC nonviable + 10^3^ VC viable	−	5.98 ± 0.68 CFU/g (VC)
	+	3.45 ± 0.46 CFU/g (VC) *
10^6^ VC nonviable + 10^2^ VC viable	−	6.72 ± 0.13 CFU/g (VC)
	+	2.15 ± 0.81 CFU/g (VC) *
10^6^ VP nonviable + 10^4^ VP viable + 10^6^ VC nonviable + 10^4^ VC viable	−	6.43 ± 0.83 CFU/g (VP)
		6.15 ± 0.38 CFU/g (VC)
	+	3.89 ± 0.22 CFU/g (VP) *
		4.28 ± 0.19 CFU/g (VC) *
10^6^ VP nonviable + 10^3^ VP viable + 10^6^ VC nonviable + 10^3^ VC viable	−	6.18 ± 0.76 CFU/g (VP)
		5.82 ± 0.51 CFU/g (VC)
	+	3.49 ± 0.29 CFU/g (VP) *
		3.38 ± 0.41 CFU/g (VC) *
10^6^ VP nonviable + 10^2^ VP viable + 10^6^ VC nonviable + 10^2^ VC viable	−	6.61 ± 0.17 CFU/g (VP)
		6.45 ± 0.32 CFU/g (VC)
	+	2.52 ± 0.75 CFU/g (VP) *
		2.44 ± 0.64 CFU/g (VC) *

**Note**: VP: *V. parahaemolyticus; VC: V. cholerae;* **: p <* 0.05.

**Table 6 foods-14-02166-t006:** The ability of PMA-qPCR to distinguish the nonviable and viable *Vibrio* cells in oysters.

Number of Infections	PMA	qPCR Quantification Results
10^6^ VP nonviable + 10^4^ VP viable	−	5.84 ± 0.68 CFU/g (VP)
	+	3.89 ± 0.47 CFU/g (VP) *
10^6^ VP nonviable + 10^3^ VP viable	−	6.13 ± 0.03 CFU/g (VP)
	+	2.98 ± 0.15 CFU/g (VP) *
10^6^ VP nonviable + 10^2^ VP viable	−	6.34 ± 0.23 CFU/g (VP)
	+	2.08 ± 0.38 CFU/g (VP) *
10^6^ VC nonviable + 10^4^ VC viable	−	6.25 ± 0.49 CFU/g (VC)
	+	4.27 ± 0.89 CFU/g (VC) *
10^6^ VC nonviable + 10^3^ VC viable	−	5.79 ± 0.38 CFU/g (VC)
	+	3.18 ± 0.95 CFU/g (VC) *
10^6^ VC nonviable + 10^2^ VC viable	−	6.16 ± 0.83 CFU/g (VC)
	+	2.27 ± 0.05 CFU/g (VC) *
10^6^ VP nonviable + 10^4^ VP viable + 10^6^ VC nonviable + 10^4^ VC viable	−	6.48 ± 0.66 CFU/g (VP)
		6.19 ± 0.24 CFU/g (VC)
	+	3.51 ± 0.67 CFU/g (VP) *
		3.83 ± 0.46 CFU/g (VC) *
10^6^ VP nonviable + 10^3^ VP viable + 10^6^ VC nonviable + 10^3^ VC viable	−	6.45 ± 0.88 CFU/g (VP)
		5.88 ± 0.37 CFU/g (VC)
	+	2.79 ± 0.23 CFU/g (VP) *
		3.58 ± 0.95 CFU/g (VC) *
10^6^ VP nonviable + 10^2^ VP viable + 10^6^ VC nonviable + 10^2^ VC viable	−	5.86 ± 0.42 CFU/g (VP)
		6.34 ± 0.28 CFU/g (VC)
	+	2.36 ± 0.19 CFU/g (VP) *
		2.27 ± 0.83 CFU/g (VC) *

**Note**: VP: *V. parahaemolyticus; VC: V. cholerae;* **: p* < 0.05.

### 3.6. Application of PMA Dual qPCR in Actual Aquatic Samples

The application of the PMA dual qPCR developed in this study was conducted in August 2017 during a sampling survey at the Shanghai Oriental International Aquatic Center. A total of 108 aquatic product samples were collected, including 48 shrimp samples, 23 shellfish samples, 18 fish samples, and 19 algae samples. The PMA dual qPCR technique was employed to survey the actual aquatic samples, with results presented in Table 7: 96 out of 108 samples tested positive for *V. parahaemolyticus*, and 24 samples tested positive for *V. cholerae*. The detection rate of *V. parahaemolyticus* was 100% in shrimp, shellfish, and fish, while it was 36.84% in algae. The detection rates of *V. parahaemolyticus* in shrimp, shellfish, fish, and algae were 14.58%, 56.52%, 16.67%, and 14.29%, respectively. The detection results of the ISO (2007) standard method were highly consistent with the PCR detection results (Table 7). The detection rate was 100%, further demonstrating the high precision of this method.

Although traditional methods (ISO standard methods) are considered the gold standard for detecting pathogenic *Vibrio* in aquatic products, they require multiple complex steps, including enrichment, selective isolation, and physiological-biochemical identification, taking up to one week to complete the entire detection process [32]. In contrast, the PMA dual qPCR technology developed in this study not only features rapid, accurate, and efficient characteristics, with the entire process from the selection of viable bacterial DNA from samples to the final PCR result output taking less than three hours (30 min of manual inoculation, 30 min of dark treatment, 30 min of light treatment, and 1 h of qPCR process), but it also selectively detects viable pathogenic *Vibrio* cells in aquatic samples, providing new technical support for the identification of *V. parahaemolyticus* hazards. In terms of monetary costs, the PMA-qPCR method has an approximate detection cost of USD 15 per sample, which is higher than the USD 5 cost per sample associated with traditional methods. Regarding labor costs, traditional culture identification methods are extremely labor-intensive, requiring specialized personnel to observe colonies for extended periods and conduct physiological and biochemical identifications, which can lead to human error. In contrast, PMA-qPCR is simple and convenient to operate, with a high degree of automation that reduces the technical requirements for personnel, thereby decreasing labor input and the risk of errors.

**Table 7 foods-14-02166-t007:** The application of PMA-qPCR in practical seafood samples.

Sample Type	Number of Samples	Number of VP Positive Samples		Number of VC Positive Samples	
		PMA-qPCR	ISO	PMA-qPCR	ISO
Shrimp	48	48	48	7	7
Shellfish	23	23	23	13	13
Fish	18	18	18	3	3
Algae	19	7	7	1	1
Total	108	96	96	24	24

**Note**: VP: *V. parahaemolyticus; VC: V. cholerae.*

## 4. Conclusions

This study developed a novel integrated PMA pretreatment instrument, integrating the two core steps of dark treatment and light crosslinking into a single process. This innovation achieves the integration and full automation of the PMA pretreatment process, thereby simplifying the handling of PMA at its source and enhancing its processing efficiency. Based on this rapid screening instrument for viable bacterial DNA, we established a quantitative detection technique for viable *V. parahaemolyticus* and *V. cholerae* in aquatic products and their associated farming environments, leveraging the advantages of multiplex qPCR. This technique demonstrates excellent amplification efficiency, low detection limits, and a strong capability to differentiate between viable and nonviable bacteria. This research provides a novel technology for accurately identifying the real hazards posed by pathogenic *Vibrio*, offering a powerful tool for the risk monitoring and assessment of pathogenic *Vibrio* in aquatic products.

## Figures and Tables

**Figure 1 foods-14-02166-f001:**
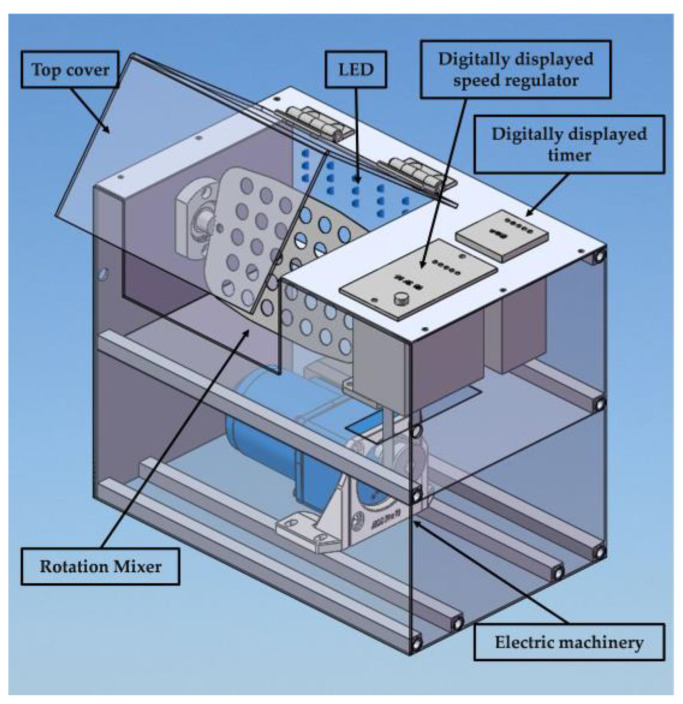
The design drawing of integrated PMA pretreatment instrument.

**Figure 2 foods-14-02166-f002:**
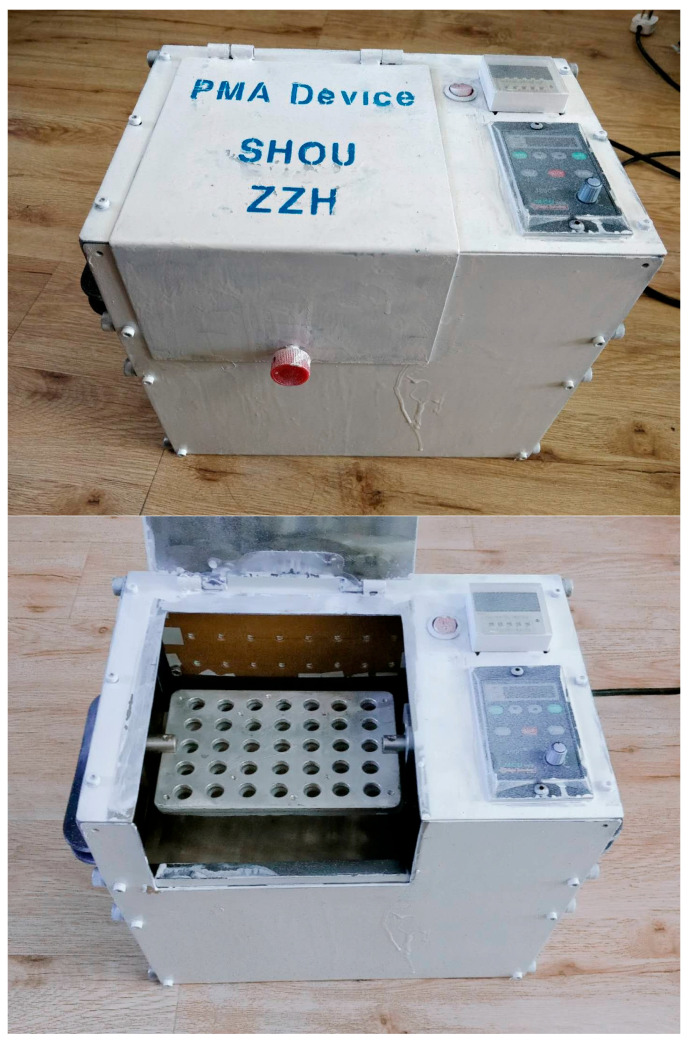
The model machine of integrated PMA pretreatment instrument.

**Figure 3 foods-14-02166-f003:**
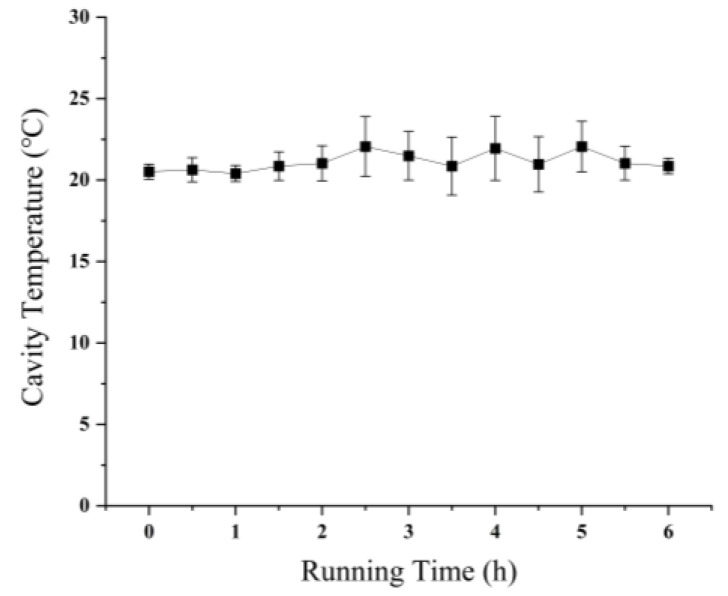
The temperature of integrated PMA pretreatment instrument as a function of time.

**Figure 4 foods-14-02166-f004:**
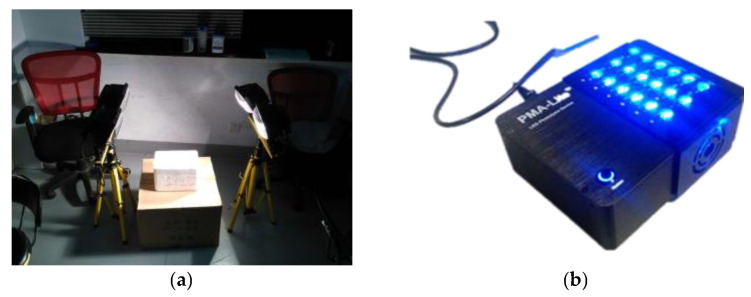
Conventional PMA treatment. (**a**) Halogen lamp; (**b**) Commercial PMA treatment device.

**Figure 5 foods-14-02166-f005:**
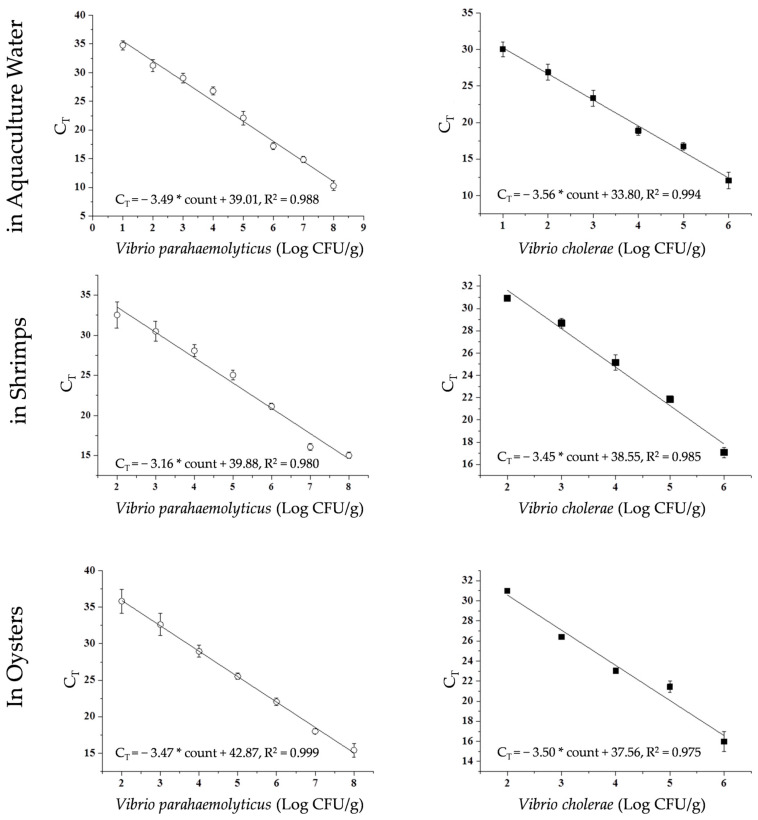
The standard curve of PMA-qPCR.

**Table 1 foods-14-02166-t001:** The primers and probes of PMA-qPCR.

Bacterium	Primers and Probes	Fluorophores	Quencher	Target Gene
VP	Forward primers: ACTCAACACAAGAAGAGATCGACAA	FAM	BHQ1	*tlh* (208 bp)
	Reverse primer: GATGAGCGGTTGATGTCCAA			
	Probes: FAM-CGCTCGCGTTCACGAAACCGT-BHQ1			
VC	Forward primers: CCGTTGAGGCGAGTTTGGTGAGA	Cy3	BHQ1	*lolB* (137 bp)
	Reverse primer: GTGCGCGGGTCGAAACTTATGAT			
	Probes: Cy3-ATGGGTTGCTTGGGTCGGCAAGCCT-BHQ1			

**Note**: VP: V. parahaemolyticus; VC: V. cholerae.

**Table 2 foods-14-02166-t002:** The bacterial strains used for specificity analysis.

Strain Species	Strain No.	Number of Strains
*Vibrio* spp.		
*V* *. parahaemolyticus*	ATCC 33847	1
	ATCC 17802	1
	VPD1-VPD20 (from seafood samples)	20
	VPC1-VPC40 (from clinical samples)	40
*V* *. cholerae*	GIM1.449	1
	VCW1-VCW30 (from clinical samples)	30
*V* *. anguillarum*	CICC 10475	1
*V* *. fluvialis*	CGMCC 1.1611	1
*V* *. vulnificus*	MCCC 1H0006	1
Other spp.		
*Listeria monocytogenes*	ATCC 19112	1
*Listeria innocua*	ATCC 33090	1
*L* *isteria welshimeri*	ATCC 43548	1
*Salmonella Enteritidis*	CMCC 50041	1
*Salmonella Typhimurium*	CICC 21484	1
*Escherichia coli* O157:H7	ATCC 43889	1
*Staphylococcus aureus*	CCTCC AB 91093	1

**Table 3 foods-14-02166-t003:** Results of PMA duplex qPCR specificity

Strain Species	Number of Strains	FAM (*tlh*)	Cy3 (*lolB*)
*V* *. parahaemolyticus*	62	+	−
*V* *. cholerae*	31	−	+
Other *Vibrio*	3	−	−
Other genera	7	−	−

**Note**: +: Positive; −: Negative

## Data Availability

The original contributions presented in the study are included in the article; further inquiries can be directed to the corresponding authors.

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
