# Peer review of "An Integrated PMA Pretreatment Instrument for Simultaneous Quantitative Detection of Vibrio parahaemolyticus and Vibrio cholerae in Aquatic Products"

_foods, 2025, doi:10.3390/foods14132166_

Round 1

Reviewer 1 Report

Comments and Suggestions for Authors

The authors present an integrated instrument for processing samples with PMA. It allows automated performance of the dark incubation step and light incubation. The development is important in the context of automation of sample analysis. The developed technology was applied to distinguish membrane-intact and membrane-compromised Vibrio parahaemolyticus and Vibrio cholerae in a duplex qPCR assay. Inclusivity was tested using an impressive number of strains of these two species. Exclusivity was also addressed with a selection of other Vibrio and non-Vibrio species.

The developed device is innovative and novel. The validation of the duplex qPCR assay would have benefited to include results of samples without PMA treatment. Potential for improvements are seen also in the way that data is presented (see specific comments below). Furthermore, data for controls should be shown to better evaluate the performance of the duplex PCR assay in terms of limit of detection.

PMA has an excitation maximum at 464 nm. Has the light dose/intensity of the LED lights in this range been quantified? Appropriate blue light LEDs emit at 460 - 465 nm.

Specific comments:

Introduction:

Line 43-44: one additional sentence about the traditional methods that require one week for detection would be adequate.

Line 54-54: PMA is not decomposed into nitrogen-rich compounds, but the azide group itself reacts and forms covalent bonds with hydrocarbons of adjacent organic molecules

Materials & Methods

Fig. 1: it might be good to indicate the position of the LED-light bulbs with dots

Line 112: what was the source of PMA? No company name is provided.

Line 115: a dark treatment time of 30 min might work for the tested Vibrio strains, but is long for other bacteria.

Table 1: the amplicons are relatively short. The amplification of longer sequences would be beneficial to improve PMA-mediated signal reduction. See Contreras et al. 2011 (Effect of PCR amplicon length on suppressing signals from membrane-compromised cells by propidium monoazide treatment).

Line 162: dead Vibrio bacteria were prepared by autoclavation. Such high temperature under pressure however leads to lysis and rupture of part of the cells. For the future it would be better to apply “milder” killing conditions that lead to killing, but not to complete rupture of bacteria.

Lines 167f: the procedure for artificial inoculation of shrimps/oysters should be described shortly.

Results & Discussion

Line 207: Biotium is not the “PMA company”. The first commercial devices for light treating PMA-incubated samples using blue LEDs were launched by the Spanish company GenIUL.

Table 3: Only +/- information is provided. What was the Cq-threshold for the FAM and Cy3 signals? This information should be stated in the Materials & Methods section. And how much DNA was used as a template for each reaction? This information should be stated.

Fig. 5: which Cq values were obtained with the negative controls, meaning from samples without Vibrio?

Tables 4-6: data is very hard to read in Table form. This reviewer recommends the presentation of data in form of a figure which allows to see the impact of the presence of heat-killed cells much more rapidly.

Tables 4-6: one misses results from samples with only heat-killed cells and no viable Vibrio bacteria.

References: check reference 18 for surnames/last names.

Reviewer 2 Report

Comments and Suggestions for Authors

Please simplify the abstract by splitting sentences into two sentences. For example line 14-18 is one sentence.

Introduction:

Line 43: Please list all traditional methods and there times for detection. Now only a statement is given, that traditional methods take 1 week, but what methods are those?

The state of the art is mostly descriptive, summarizing how PMA works and the issues with current methods. It does not critically compare alternative methods (e.g., viability PCR variants, alternative dyes, or non-PCR-based methods). There is also no discussion of performance metrics, such as sensitivity, specificity, reproducibility, or cost, which are essential for evaluating method effectiveness. Furthermore the state of the art focuses heavily on PMA, without discussing other dyes (like EMA – ethidium monoazide) or advanced viability detection methods (e.g., RNA-based techniques, flow cytometry).

This may unintentionally bias the reader toward one solution.

Material and methods

For qPCR informations are missed. Please refer to the MIQE guidelines (https://pubmed.ncbi.nlm.nih.gov/19246619/) to list all informations.

The CFU concentrations (e.g., 101 to 108) are mentioned without specifying how many replicates were done or whether they were done in triplicate, which is standard practice for statistical reliability.

There’s no mention of an NTC (a standard PCR/qPCR control without any DNA), which is essential to rule out contamination in reagents or during setup.

There's no indication that an internal extraction control (e.g., spike-in DNA or a housekeeping gene) was used to ensure DNA was effectively extracted in each sample.

While mixed and non-viable inocula were tested, the design lacks an explicit PMA-treated viable-only sample group to confirm that PMA does not inhibit viable bacterial DNA amplification.

There's no reference to the use of positive control DNA or cells with a known quantity for qPCR calibration or standardization outside the standard curve step.

Results:

No formal statistical tests (e.g., t-tests, ANOVA) are reported to determine whether differences between PMA-qPCR and traditional qPCR are statistically significant.

Error margins (±) are provided, but no indication of sample size (n) or statistical power is given.

There’s redundancy and inconsistency in how results are reported: Sometimes “CFU/mL,” sometimes “CFU/g”.Some values are shown with 2 decimals, others with 3.

The claim that the method ensures "high safety and operational efficiency" or that it "can serve as a novel hazard identification technology" is overstated in my opinion unless it has been tested in a validated industrial setting and there’s regulatory benchmarking beyond ISO comparison.

Claims of high sensitivity and specificity are not quantified (e.g., no ROC curve, PPV, NPV, false-positive/negative rates).

No exact time comparisons are given (e.g., total minutes/hours from sample to result). What is the benefit?

No cost analysis or resource requirements (equipment, PMA cost, sample throughput) are described.

It lacks any performance comparison with commercial test kits or molecular platforms.

It’s not shown whether the assay cross-reacts with other Vibrio species (e.g., V. vulnificus, V. alginolyticus) or unrelated bacteria commonly found in seafood.

Round 2

Reviewer 1 Report

Comments and Suggestions for Authors

The authors have developed an innovative device for PMA treatment. All points raised by the reviewer have been addressed. The presentation of data can be improved in the future using more visual diagrams. 

Author Response

Thank you for your insightful comments. We're glad that our innovative PMA treatment device has caught your attention. We'll take your suggestion on board and enhance data visualization with more diagrams in future revisions.

Reviewer 2 Report

Comments and Suggestions for Authors

Dear authors,

thank you for presenting this timely and technically innovative manuscript. The integration of PMA pretreatment and duplex qPCR for detecting Vibrio parahaemolyticus and Vibrio cholerae in seafood is certainly relevant and addresses a real need in the field. That said, there are several important issues that require your attention to ensure both scientific accuracy and methodological transparency.

The current framing of detection approaches in your introduction is too narrowly focused on cultivation-based methods. While you rightly note the limitations of classical culture techniques, the omission of widely used alternatives creates a biased impression. Please consider including:

  • Immunological methods (e.g., ELISA, Latex Agglutination)
  • Lateral flow assays, which are especially relevant in field settings and food testing.

Including these will provide a balanced state-of-the-art and better contextualize your contribution.

MIQE Guidelines

Although you have addressed some reviewer comments, it is essential to take a comprehensive look at the MIQE guidelines, rather than responding to selected points. Below is a summary of key missing or insufficiently reported data, which must be addressed – beside others!:

  • No details on DNA quantification (e.g., spectrophotometric ratios, fluorometric concentration) or purity are provided.
  • While sequences are included, no information is given on in silico specificity checks (e.g., BLAST), or melt curve analysis.
  • Especially in complex matrices like shrimp and oyster, the use of internal amplification controls or spiked samples to test for inhibition is critical—yet completely absent.
  • LOD vs. LOQ: The manuscript estimates LOD, but no statistical method is described, and limit of quantification (LOQ) is not mentioned.
  • NTCs are reported, but there is no mention of mock extraction controls or No Amplification Controls (NACs), which are essential for verifying assay specificity and contamination risk.
  • Though experiments are said to be in triplicate, there is no presentation of raw CT values, no variation statistics (SD or CV), and no mention of technical vs. biological replicates.

Let me emphasize: these omissions are not minor. They significantly impact the credibility and reproducibility of your qPCR-based method and must be addressed in full.

Figure 3: Although this figure lacks essential statistical information. Please indicate how many replicates were performed and add appropriate error bars to support your temperature stability claim.

Round 3

Reviewer 2 Report

Comments and Suggestions for Authors

Thanks for revision of all points. I have no further comments.